# Antioxidant and Anticholinesterase Activities of *Macrosphyra Longistyla* (DC) Hiern Relevant in the Management of Alzheimer’s Disease

**DOI:** 10.3390/antiox8090400

**Published:** 2019-09-16

**Authors:** Taiwo O. Elufioye, Chidimma G. Chinaka, Adebola O. Oyedeji

**Affiliations:** 1Department of Chemistry, Walter Sisulu University, Mthatha 5117, South Africa; aoyedeji@wsu.ac.za; 2Department of Pharmacognosy, Faculty of Pharmacy, University of Ibadan, Ibadan 200284, Nigeria; chinakachidmma@gmail.com

**Keywords:** *Macrosphyra longistyla*, cholinesterase, antioxidant, total phenolic, total flavonoid

## Abstract

*Macrosphyra longistyla* has been used in many traditional systems of medicine for its anti-hemorrhagic, antidiabetic, anti-ulcer, and anti-diarrhea properties. The acetylcholinesterase (AChE) and butyrylcholinesterase (BuChE) inhibitions of the crude methanol extracts and its various partitioned fractions were determined by a modified method of Ellman. An evaluation of the antioxidant activity was carried out using 1,1-diphenyl-2-picryl-hydrazyl (DPPH) radical scavenging, ferric reducing power, and nitric oxide scavenging assays. The total flavonoids were estimated based on the aluminum chloride method, while the total tannins and phenolics were estimated based on the vanillin–HCl and Folin–Ciocalteu method, respectively. The ethyl acetate fraction had the highest DPPH radical scavenging activity, and the highest ferric reducing power with a concentration providing 50% inhibition (IC_50_) of 0.079 mg/mL and 0.078 mg/mL, respectively, while the crude methanol extract had the highest nitric oxide scavenging activity with an IC_50_ of 0.008 mg/mL. The methanol extract had the highest phenolics and flavonoids contents, while the aqueous fraction had the highest tannin content. The crude methanol extract had the best AChE and BuChE inhibitory action, with an IC_50_ of 0.556 µg/mL and 5.541 µg/mL, respectively, suggesting that the plant had a better AChE inhibiting potential. A moderate correlation was observed between the phenolic content and DPPH radical scavenging, NO radical scavenging, and AChE inhibitory activities (*r^2^* = 0.439, 0.430, and 0.439, respectively), while a high correlation was seen between the flavonoid content and these activities (*r^2^* = 0.695, 0.724, and 0.730, respectively), and the ferric reducing antioxidant power correlated highly with the proautocyanidin content (*r^2^* = 0.801). Gas chromatography mass spectrometry (GCMS) revealed decanoic acid methyl ester (24.303%), 11,14-eicosadienoic acid methyl ester (16.788%), linoelaidic acid (10.444%), pentadecanoic acid (9.300%), and 2-methyl-hexadecanal (9.285%). Therefore, we suggest that *M. longistyla* contain bioactive chemicals, and could be a good alternative for the management of Alzheimer’s and other neurodegenerative diseases.

## 1. Introduction

Alzheimer’s disease and other neurodegenerative conditions are usually characterized by the slow, but progressive, dysfunction and loss of neurons in the central nervous system [1]. About 55 million people are suffering from one form of neurodegenerative disease (ND) or another, with an expected rise in this figure the with increasing age of the population [2,3]. Despite the volume of research on the pathogenesis of neurodegenerative conditions, appropriate treatment is yet to be found [4]. However, several factors, including aging [5] and some pathological conditions, such as impaired mitochondrial function [6], aggregated proteins deposit [7], neuroinflammation [8], cholinergic deficit [9], and oxidative stress [10], have been associated with NDs. Thus, the management of NDs involves addressing one or more of the associated conditions. The currently available therapies for Alzheimer’s disease (AD) are cholinesterase inhibitors such as rivastigmine and donepezil, which only reduce disease progression and provide symptomatic relieve [11]. Thus, efforts are still being made to find alternative and better therapeutic options.

*Macrosphyra longistyla* is a shrub found in several tropical countries. It has long, arching stems that are about 4 m long [12]. *M. longistyla* has been used traditionally as an antihemorrhagic in Benin [13], as an antidiabetic in Nigeria and Côte d’Ivoire [14,15], as a contraceptive and for the restoration of fertility [16], for ulcers [17], and for diarrhea [18]. Fresh wildly-collected leaves are eaten as a vegetable by the Gourmantché, Aïzo, and Cotafon people in Benin [19]. It is also widely consumed in Togo [20]. The leaves have been suggested an indigenous food ingredient for complementary food formulations to combat infant malnutrition [21]. To the best of our knowledge, the chemical constituents and biological activity of this plant have not been reported in the literature. This study therefore investigates its anticholinesterase and anti-oxidant potentials, as well as phytochemical characterization.

## 2. Materials and Methods

### 2.1. Plant Material

The leaves of *Macrosphyra longistyla* were collected from Agbogi village in Osun State in December 2017. The plant was identified and authenticated by Mr. Odewo of the Forest Herbarium Ibadan (FHI), with voucher number FHI 112042. The voucher specimen were deposited at the herbarium of the Department of Pharmacognosy, University of Ibadan.

### 2.2. Plant Extraction and Partitioning

The leaves were air-dried and pulverized. About 2.25 kg of the powdered leaf was macerated using 100% methanol. The extract was filtrated using a Buchner funnel, and concentrated in vacuo so as to obtain a crude methanol extract. Then, 80 g of the crude methanol extract was partitioned into *n*-hexane, ethyl acetate, and water, to obtain the respective fractions, which were concentrated in vacuo and used for the subsequent experiments. The percentage yield of both extract and fractions were determined.

### 2.3. Phytochemical Screening

The preliminary phytochemical screening of the crude methanol extract was carried out using standard procedures. These include tests for alkaloids using Dragendorff and Wagner reagents, the Borntrager’s test for anthraquinones, and a ferric chloride test for phenolic compounds [22,23].

### 2.4. Determination of the Total Phenol Content (TPC)

The total phenol content in the methanolic extract and various fractions of *M. longistyla* were determined based on a previously described procedure [24]. Then, 2.5 mL of 10% Folin–Ciocalteau’s reagent was mixed with 2 mL of 2% sodium carbonate solution (Na_2_CO_3_), followed by the addition of 0.5 mL of methanolic extract and fractions of *M. longistyla* (1 mg/mL). The mixture was incubated at 45 °C for 15 min, and absorbance was taken at 765 nm. The quantification was done with respect to the standard of gallic acid at different concentrations (1, 0.5, 0.25, 0.125, 0.063, and 0.031 mg/mL). The content of the total phenolic compounds was calculated based on a standard curve prepared using gallic acid and expressed as milligrams of gallic acid equivalent (GAE) per gram of sample.

### 2.5. Determination of Total Flavonoid Contents (TFC)

The total flavonoid content was determined using the aluminum chloride colorimetric method [25]. In this method, 1 mL of crude extract or fractions of *M. longistyla* were mixed with 3 mL of methanol, followed by 0.2 mL of 10% aluminum chloride (AlCl_3_), 0.2 mL of potassium acetate (1 M), and 5.6 mL of distilled water, and left at room temperature for 30 min. Absorbance was taken at 420 nm. Quantification was done with respect to the standard of gallic acid at different concentrations (1, 0.5, 0.25, 0.125, 0.063, and 0.031 mg/mL). The total phenolic content was calculated based on a standard curve prepared using gallic acid, and expressed as milligrams of gallic acid equivalent (GAE) per gram of sample.

### 2.6. Determination of Pro-Anthocyanidin Content (PAC)

The vanillin–HCl method was used for the quantitative determination of condensed tannins (proanthocyanidins) [26]. In this method, 3 mL of 4% vanillin in methanol, and 1.5 mL of hydrochloric acid (HCl) was added to 0.5 mL of extract/fractions (1 mg/mL). The mixture was vortexed thoroughly and allowed to stand for 15 min at room temperature. Absorbance was read at 500 nm. A calibration curve was prepared using a standard gallic acid solution. All of the results were expressed as mg gallic acid equivalents (GAE) per gram of sample.

### 2.7. DPPH (2,2-Diphenyl-1-Picrylhydrazyl Hydrate) Radical Scavenging Assay

The radical scavenging ability of the fractions was determined using the stable radical DPPH (2,2-diphenyl-1-picrylhydrazyl hydrate), as previously described [27]. In this assay, 1 ml of 0.1 mM DPPH was mixed with 1 mL of crude extract and fractions of *M. longistyla* at different concentrations (1, 0.5, 0.25, 0.125, 0.063, and 0.031 mg/mL), as well as the positive controls (ascorbic acid and 2,6-di-tert-butyl-4-methylphenol (DDM)) at different concentrations (1, 0.5, 0.25, and 0.125 mg/mL). The reaction was vortexed and left in the dark at room temperature for 30 min, after which the absorbance was taken at 517 nm. The percentage inhibition was calculated as follows:I% = [(A_blank_ − A_sample_)/A_blank_] × 100
where A_blank_ is the absorbance of the control reaction (containing all reagents except the test compound), and A_sample_ is the absorbance of the test compound. The sample concentration providing 50% inhibition (IC_50_) was also calculated.

### 2.8. Nitric Oxide (NO) Scavenging Assay

The nitric oxide scavenging assay was carried out as previously described [28]. First, 2 mL of sodium nitroprusside was mixed with 0.5 mL of phosphate buffer pH 7.4 and 0.5 mL of different concentrations of extract (0.0031–1.0 mg/mL). The mixture was incubated at 25 °C for 150 min, and an initial absorbance (A0) was taken at 540 nm. Thereafter, 0.5 mL of the incubated mixture was mixed with 1 mL of a sulfanilic acid reagent and 1 mL of naphthylethylenediamine dichloride (0.1% *w*/*v*), and incubated at room temperature for 30 min, before another absorbance (A1) was taken at 540 nm. The same reaction mixture without the extract but with the equivalent amount of methanol served as the negative control. Ascorbic acid and DDM at various concentrations were used as the standard. All of the experiments were in triplicates. The percentage nitrite radical scavenging activity of the extracts and standard were calculated using the following formula:% inhibition of NO = [A0 − A1]/A0 × 100
where A0 is the absorbance before the reaction, and A1 is the absorbance after the reaction.

### 2.9. Ferric Reducing Antioxidant Assay

The reducing power was determined according to the method of Oyaizu [29]. Substances with a reducing ability react with potassium ferricyanide (Fe^3+^) to form potassium ferrocyanide (Fe^2+^), which then reacts with ferric chloride to form a ferric ferrous complex that has an absorption maximum at 700 nm. Briefly, 0.2 mL of various concentrations of plant extract and fractions was mixed with 0.2 mL of phosphate buffer and 0.2 mL of potassium ferricyanide. The mixture was vortexed and incubated at 50 °C for 20 min. After cooling, 0.2 mL of 10% trichloroacetic acid (TCA) was then added to the mixture and centrifuged at 4500 rpm for 10 min. Then, 100 µL of the upper solution was mixed with 20 µL of the ferric chloride solution and 100 µL of distilled water. The absorbance was taken at 700 nm. The control was prepared in a similar manner, but without the test sample. Ascorbic acid and DDM at various concentrations were used as the standard. The experiments were done in triplicates.

### 2.10. Cholinesterase Inhibitory Assay

Acetylcholinesterase (AChE) and butyrylcholinesterase (BuChE) inhibitions were determined spectrophotometrically using acetylcholine iodide and butrrylcholine iodide as substrates, respectively, by a modified method of Ellman [30]. The serial dilutions of the fractions were subjected to this test using eserin and donepezil as the positive control. Then, 5 mg of both the extract and fractions were dissolved in 1 mL of methanol. Serial dilutions of each sample were done in order to obtain the final concentrations of 1, 0.5, 0.25, 0.125, and 0.0625 mg/mL, while the positive controls (eserin and donepezil) were also diluted serially to obtain the final concentrations of 0.1, 0.05, 0.025, 0.0125, 0.00625, and 0.003125 mg/mL. Thereafter, 20 µL of each concentration was pipetted into the micro plates, followed by 240 µL of the phosphate buffer (pH 8) and 20 µL of the enzyme, which was then vortexed. The plates were then incubated at 37 ºC for 30 min. After incubation, 20 µL of 25 mM of the substrate (acetylthiocholinecholine iodide (ATChI) or butyrylthiocholine chloride (BTChCl)) was added to the reaction mixture, followed by the addition of 20 µL of 10 mM 5, 5ʹ-Dithiobis-2-nitrobenzoic acid (DTNB). The hydrolysis of acetylcholine iodide or butryrylthiocholine chloride was determined spectrophotometrically at 412 nm. The assay was carried out in triplicates, with methanol as the negative control. The percentage inhibition was computed using the following formula:$$\frac{\Delta a-\Delta b}{\Delta a}\times 100$$
where Δ*a* is the change in absorbance of the negative control, and Δ*b* is the change in absorbance of the sample.

### 2.11. Gas Chromatography Mass Spectrometry (GCMS) Analysis

One microliter (1 µL) of the sample diluted in hexane was analyzed on a Bruker 450 gas chromatography-300 mass spectrometer (GCMS) system operating in EI mode at 70 eV, equipped with a HP-5 MS fused silica capillary system with a 5% phenylmethylsiloxane stationary phase. The capillary column parameter was 30 m by 0.25 mm, while the film thickness was 0.25 µm. The initial temperature of the column was set at 70 °C, and heated to 240 °C at a rate of 5 °C/min, with the final temperature kept at 450 °C. The run time was 66.67 min, and helium was used as the carrier gas at a flow rate of 1 min/min. The split ratio was 100:1. The scan time was 78 min, with a scanning range of 35 to 450 amu.

### 2.12. Statistical Analysis

All of the data were analyzed using GraphPad Prism 6.0, and were expressed as mean ± standard error of the mean (SEM). The correlation and regression analysis of the activities (Y) versus the total phytochemical content (X) were carried out using the online Quest Graph™ Linear, Logarithmic, Semi-Log Regression Calculator [31].

## 3. Results and Discussion

*M. longistyla* has been reportedly used for managing different ailments in traditional medicine [13,14,15,16,17,18]. In an ethnomedical survey carried out by us, the plant was mentioned as a memory enhancer. Thus, the present study was carried out to investigate its phytochemical content, as well as evaluate the antioxidant and cholinesterase inhibitory activities of the extracts and partitioned fractions.

The preliminary phytochemical screening of the methanol extract revealed the presence of tannins, flavonoids, phenolics, terpenoids, and saponins. Anthraquinones and alkaloids were, however, found absent in the plant (Table 1).

The percentage yield of the extract and fractions (expressed as weight of extract/fraction relative to the weight of the initial plant material) ranged from 4.70% to 40.00%, with the highest being the aqueous fraction (Table 2). This suggests that the polar solvent was able to extract more constituents, probably because of the solubility of the polar compounds present in the plant material.

Furthermore, the content of the phenols (TPC), flavonoids (TFC), and the tannins (PAC) was estimated quantitatively. The TPC, as determined by the Folin–Ciocalteu method, ranged from 7.56 ± 0.12 to 18.30 ± 0.04 mg GAE/g of extract (Table 2). Both the crude extract and the various fractions had an appreciable total phenolic content, with the methanol extract and the ethyl acetate fractions having the highest TPC, while the *n*-hexane fraction had the least TPC. The total flavonoid and proauthocyanidin contents, also reported as mg GAE/g of extract, showed that the TFC ranged from 5.02 ± 0.01 to 16. 07 ± 0.14 mg GAE/g of extract, while the PAC ranged from 2.99 ± 0.06 to 26.11 ± 0.02. In both cases, the hexane fraction had the least amount (Table 2).

Phenolic compounds are present in plant tissues and serve as antioxidants [32], because of the presence of hydroxyl groups, which are responsible for their scavenging ability. Thus, they are capable of reacting with active oxygen radicals such as hydroxyl radicals [33]. Flavonoids are polyphenolic compounds, and are responsible for some of the health benefits of vegetable and fruits [34]. They are known to play an active role in the quenching of free radicals, because of their redox properties [35]. Tannins, however, are a high molecular weight polyphenolic that have also been implicated as antioxidants [36].

The antioxidant activity of the extract and fractions was evaluated by the DPPH and NO radical scavenging activity, as well as the ferric reducing power, while the AChE inhibitory activity was evaluated by Ellman’s colorimetric assay.

DPPH is usually reduced by a hydrogen donating compound, leading to its change in color, from deep violet to light yellow, which can be monitored spectrophotometrically [37]. The DPPH radical scavenging activity results are as shown in Figure 1, while the IC_50_—the concentration of antioxidant (extract/fractions) required for 50% scavenging of DPPH radicals—values are given in Table 3. From the results, the ethyl acetate fraction had the highest activity, with an IC_50_ value of 0.078.

Nitric oxide is important in the regulation of several physiological processes, and several diseases have been associated with a high concentration of NO [38]. The nitric oxide scavenging activity can be determined by estimating for nitrate and nitrite, using the Greiss Illosvoy reaction [39]. At a physiological pH (7.2), sodium nitroprusside decomposes in an aqueous solution to produce NO, which reacts with oxygen to form stable products—nitrate and nitrite. Scavengers of NO compete with oxygen, leading to a reduced production of nitrite ions [40].

In the nitric oxide scavenging assay, all of the extract and fractions exhibited a good scavenging effect, with the methanol extract having the best scavenging effect (IC_50_ = 0.008), followed by an aqueous fraction (IC_50_ = 0.010) and then the ethyl acetate fraction (IC_50_ = 0.056; Figure 2 and Table 3).

Ferric reducing power is well linked with antioxidant activity [41], and compounds with a reducing effect are usually electron donors that can reduce oxidized intermediates of lipid peroxidation processes, thus acting as primary or secondary antioxidants [33]. In the ferric reducing antioxidant assay, the methanol extract, aqueous fraction, and the ethyl acetate fraction had good reducing activity (Figure 3), with IC_50_ values of 0.051, 0.009, and 0.078 respectively.

On the whole, a better antioxidant activity was observed in the polar fractions, and this could be because of the abundant presence of major secondary metabolites, such as tannins and flavonoids, in these fractions, as supported by the higher TPC and TFC in these fractions. Phenolics are free-radical terminators [33], thus having protective effects against many infectious and neuro degenerative diseases such AD [42].

The inhibition of cholinesterase enzymes is considered promising in the management of neurological and neurodegenerative disorders such as AD, senile dementia, ataxia, and myasthenia gravis, where a deficit in cholinergic neurotransmission is often observed [43,44]. Compounds with a dual inhibitory effect on AChE and BuChE are also considered better, as BuChE also plays a minor role in the regulation of AChE [45,46].

In this study, the methanol extract inhibited the acetylcholinesterase enzyme the most, followed by the ethyl acetate and aqueous fractions, with respective percentage inhibitions of 81.629 ± 0.02, 76.985 ± 0.04, and 71.778 ± 0.01 (Figure 4). The hexane fraction had the least inhibitory action, suggesting that the active constituents are likely to be polar. The study also suggests a better inhibition of AChE as compared to BuChE, as both the crude extract and the various fractions had a lower percentage inhibition and higher IC_50_ values in the later enzyme (Figure 5 and Table 4).

The cholinesterase inhibitory activity of several medicinal plants has been reported in the literature [47,48,49,50,51]. Also, antioxidants such as vitamin E and vitamin C have been reportedly associated with a decrease in AD incidence and prevalence, [52] and AD patients on high doses of antioxidants were reported to have a slower rate of cognitive deterioration [53]. Thus, the good antioxidant and anticholinesterase activities of polar fractions in this study suggest that these fractions are good sources of phenolic compounds, with potential cholinesterase inhibitory and antioxidant properties that may find usefulness in the management of AD. This is the first report of such activities in *Macrosphyra longistyla.*

We also correlated the phytochemical content with the observed activities of the plant. Several pharmacological effects of the plant extract such, as being anti-inflammatory, antioxidant, and antimicrobial, have also been associated with the presence of phenolic compounds [54,55], and *r^2^* values have been used to show the relationship between the phytochemical constituents and activities of medicinal plants [56]. There was a moderate correlation between the total phenolic content and the DPPH and NO radical scavenging, as well as the AChE inhibitory activities (*r^2^* = 0.439, 0.430, and 0.439, respectively). However, a better correlation was observed between the flavonoid content and these activities (*r^2^* = 0.695, 0.724, and 0.730, respectively), while the ferric reducing antioxidant power correlated with the proautocyanidin content (*r^2^* = 0.801; Table 5).

Finally, the identification of possible compounds in the non-polar (hexane) fraction using GC-MS revealed the presence of twenty-three compounds (Table 6). The most abundant was decanoic acid methyl ester (24.303%), followed by 11,14-eicosadienoic acid methyl ester (16.788%), linoelaidic acid (10.444%), pentadecanoic acid (9.300%), and 2-methyl-hexadecanal (9.285%).

## 4. Conclusions

This study revealed the antioxidant and anticholinesterase activities of the compounds present in *M*. *longistyla*, and suggest the potential use of extracts from this plant for the management of neurodegenerative conditions. The polar fractions had the highest antioxidants and anticholinesterase constituents, which can be further exploited. Also, the GCMS analysis identified the compounds likely to contribute to the observed activities.

## Figures and Tables

**Figure 1 antioxidants-08-00400-f001:**
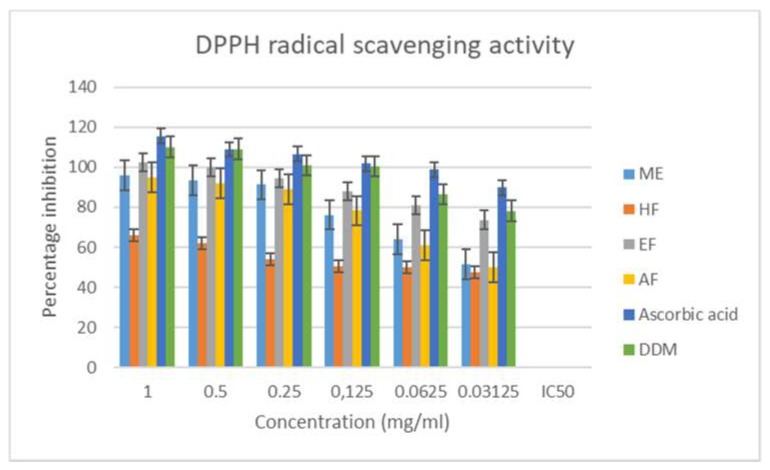
1,1-diphenyl-2-picryl-hydrazyl (DPPH) radical scavenging activity of extract and fractions of *M. longistyla.* Data are expressed as mean ± standard deviation (SD; *n* = 3). ME—methanol extract; HF—hexane fraction; EF—ethyl acetate fraction; AF—aqueous fraction; DDM—2,6-di-tert-butyl-4-methylphenol.

**Figure 2 antioxidants-08-00400-f002:**
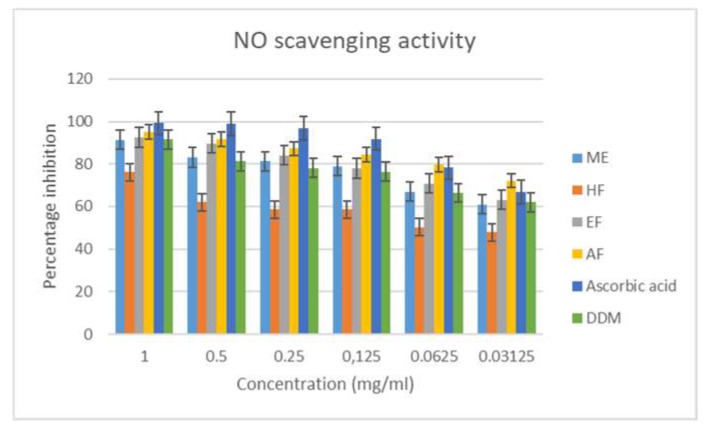
Nitic oxide (NO) radical scavenging activity of the extract and fractions of *M. longistyla.* Data are expressed as mean ± SD (*n* = 3). ME—methanol extract; HF—hexane fraction; EF—ethyl acetate fraction; AF—aqueous fraction; DDM—2,6-di-tert-butyl-4-methylphenol.

**Figure 3 antioxidants-08-00400-f003:**
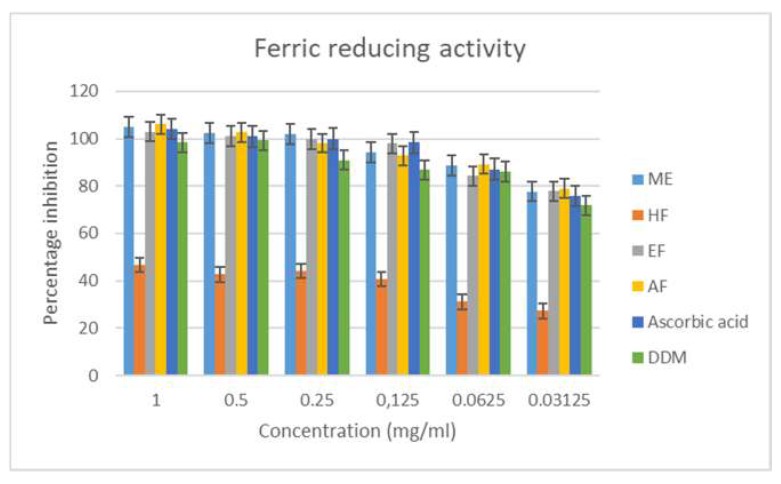
Ferric reducing activity of the extract and fractions of *M. longistyla.* Data are expressed as mean ± SD (*n* = 3). ME—methanol extract; HF—hexane fraction; EF—ethyl acetate fraction; AF—aqueous fraction; DDM—2,6-di-tert-butyl-4-methylphenol.

**Figure 4 antioxidants-08-00400-f004:**
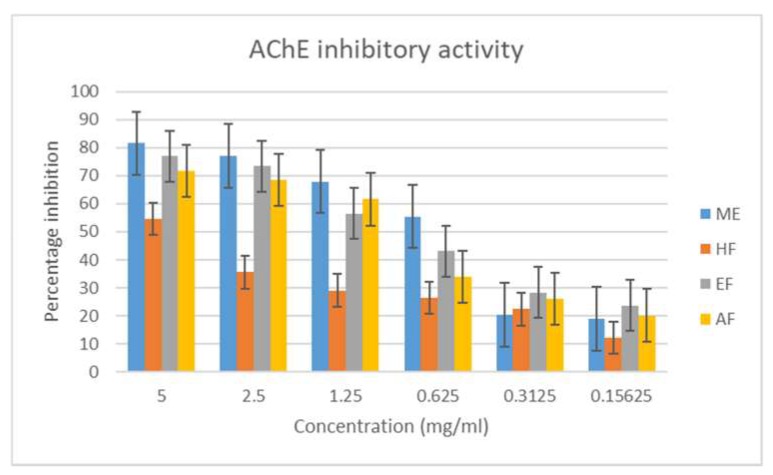
Acetylcholinesterase inhibitory activity of the extract and fractions of *M. longistyla.* Data are expressed as mean ± SD (*n* = 3). ME—methanol extract; HF—hexane fraction; EF—ethyl acetate fraction; AF—aqueous fraction.

**Figure 5 antioxidants-08-00400-f005:**
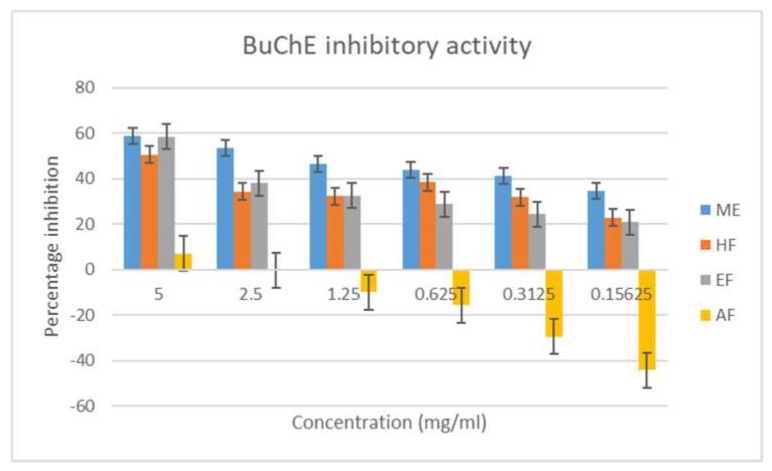
Butryrylcholinesterase inhibitory activity of the extract and fractions of *M. longistyla.* Data are expressed as mean ± SD (*n* = 3). ME—methanol extract; HF—hexane fraction; EF—ethyl acetate fraction; AF—aqueous fraction.

**Table 1 antioxidants-08-00400-t001:** Phytochemical screening results of *Macrosphyra longistyla.*

Tests	Observations	Inferences
1. Alkaloids		
a. Dragendorff	Deep yellow color	Alkaloid absent
b. Wagner test	Orange color	Alkaloid absent
2. Anthraquinones		
a. Borntrager’s test	Milky color	Anthraquinone absent
3. Flavonoids	Yellow coloration	Flavonoids present
4. Phenols	Dark coloration	Phenols present
5. Tannins	Blue black coloration	Tannin present
6. Saponin	Frothing which disappear after sometime	Saponin present
7. Terpenoid	Dark green coloration	Terpenoids present

**Table 2 antioxidants-08-00400-t002:** The total phenolics, flavonoids, and authocyanidins content in the extract and fractions of *M. longistyla.*

Assays	ME	HF	EF	AF
% Yield	6.18	4.70	7.11	40.00
Total phenolics (mg GAE/g)	18.30 ± 0.04	7.56 ± 0.12	16.06 ± 0.13	9.02 ± 0.02
Total flavonoids (mg GAE/g)	16. 07 ± 0.14	5.02 ± 0.01	10.49 ± 0.014	11.62 ± 0.01
Total tannins (mg GAE/g)	24. 44 ± 0.32	2.99 ± 0.06	9.12 ± 0.17	26.11 ± 0.02

Data are expressed as mean ± standard error of the mean (SEM; *n* = 3). GAE—gallic acid equivalent; ME—methanol extract; HF—hexane fraction; EF—ethyl acetate fraction; AF—aqueous fraction.

**Table 3 antioxidants-08-00400-t003:** The concentration providing 50% inhibition (IC_50_) values of the different antioxidant assays.

Assays	IC_50_					
	ME	HF	EF	AF	Ascorbic Acid	DDM
DPPH scavenging	0.090	0.363	0.079	0.089	0.006	0.050
NO scavenging	0.008	5.678	0.056	0.010	0.072	0.063
Ferric reducing	0.051	0.087	0.078	0.009	0.053	0.003

ME—methanol extract; HF—hexane fraction; EF—ethyl acetate fraction; AF—aqueous fraction; DDM—2,6-di-tert-butyl-4-methylphenol; DPPH—1,1-diphenyl-2-picryl-hydrazyl; NO—nitric oxide.

**Table 4 antioxidants-08-00400-t004:** IC_50_ values for the cholinesterase inhibitory assay. AChE—acetylcholinesterase; BuChE—butyrylcholinesterase.

Assays	IC_50_					
	ME	HF	EF	AF	Eserin	Donepezil
AChE	0.556	25.871	0.914	0.846	0.002	0.001
BuChE	5.541	11.957	23.338	ND	0.002	0.001

ND: Not determined.

**Table 5 antioxidants-08-00400-t005:** Correlation of the total phenolic, total flavonoid, and proautocyanidin contents with antioxidant and anticholinesterase activities.

Assays	*r^2^* Values		
	Total Phenolics	Total Flavonoids	Proautocyanidin
**DPPH scavenging**	0.439	0.695	0.515
**NO scavenging**	0.430	0.724	0.558
**Ferric reducing**	0.012	0.276	0.801
**AChE inhibition**	0.439	0.730	0.557
**BuChE inhibition**	0.00154	0.131	0.325

**Table 6 antioxidants-08-00400-t006:** Compounds identified through gas chromatography mass spectrometry (GCMS).

S/N	Name of Identified Compounds	Retention Time (min)	% Abundance	Molecular Formula	Class of Compound	Reported Biological Effect	References
1	2,6,8-trimethyl-decane	27.084	0.084	C_13_H_28_	Alkane	Antifungal	[57]
2	2-methyl-hexadecanal	30.833	9.285	C_17_H_34_O	Aldehyde	Antifungal	[58]
3	Z,Z,Z-1,4,6,9-nonadecatetraene	34.147	0.349	C_19_H_32_	Alkene	Antioxidant	[59]
4	2-dodecanone	34.831	0.357	C_12_H_24_O	Aliphatic ketones	Nematocidal	[60]
5	2-pentadecanone	35.591	1.630	C_15_H_30_O	Ketone	Cytotoxic and repellant	[61,62]
6	17-octadecanal	39.211	0.119	C_18_H_36_O	Long-chain aldehyde	NR	NR
7	Hexadecanoic acid	39.564	0.152	C_16_H_32_O_2_	Saturated fatty acid	Anticancer and anthelmintic	[63,64]
8	2-methyl-dodecanoic acid	43.071	0.283	C_11_H_22_O_2_	Fatty acid	Antimicrobial	[65]
9	Neophytadiene	43.482	0.109	C_20_H_38_	Sesquiterpene	Anti-inflammatory	[66]
10	2-nonadecanone	43.611	1.942	C_19_H_38_O	Alkanone	Antimicrobial	[67]
11	Decanoic acid methyl ester	46.414	24.303	C_11_H_22_O_2_	Fatty acid ester	Antimicrobial	[68]
12	Phytol	47.053	0.202	C_20_H_40_O	Diterpene alcohol	Antinociceptive, antioxidant, and anticholinesterase	[69,70]
13	Eicosanoic acid ethyl ester	48.585	5.265	C_22_H_44_O_2_	Fatty acid	Anticancer	[71]
14	Pentadecanoic acid	48.592	9.300	C_15_H_30_O_2_	Saturated fatty acid	Anthelmintic	[64]
15	tetradecanoic acid-12-methyl-methyl ester	49.580	0.048	C_16_H_32_O_2_	Fatty acid	Anticancer and antifungal	[72,73]
16	11,14-eicosadienoic acid methyl ester	51.562	16.788	C_21_H_38_O_2_	Fatty acid	Antioxidant and anti-amylase	[74]
17	8,11,14-ecosatrienoic acid	51.712	2.299	C_20_H_34_O_2_	Omega fatty acid	Atopic dermatitis and malignant hypertension	[75,76]
18	*Z*-methyl-hexadec-11-enoate	51.963	4.204	C_17_H_32_O_2_	Fatty acid methyl ester	Antimicrobial	[77]
19	Dodecanoic acid-10-methyl-methyl ester	52.643	1.957	C_14_H_28_O_2_	Fatty acid methyl ester	Anticoagulant	[78]
20	Linoelaidic acid	53.559	10.444	C_18_H_32_O_2_	Omega-6 trans fatty acid	Anticholinesterase, anti-mycobacterium, anticancer, and antioxidant	[79,80,81,82]
21	*Z,E*-3,13-octadecadien-1-ol	53.763	2.707	C_18_H_34_O	Fatty alcohol	Antimicrobial	[83]
22	(*Z*)-methyl-Heptadec-9-enoate	53.953	0.179	C_18_H_34_O_2_	Fatty acid	Antibiotic	[84]
23	Hexadecanoic acid-2-methyl-methyl ester	54.615	1.139	C_18_H_36_O_2_	Fatty acid methyl esters	Antimicrobial and antioxidant	[85]

NR: Not reported. The various identified compounds have been reported to have different biological effects, such as being antimicrobial, antioxidant, anticoagulant, anticholinesterase, anticancer, and anthelmintic. All of these ultimately contribute to the overall activity of the plant.

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
