# Peer review of "Antioxidant and Anticholinesterase Activities of Macrosphyra Longistyla (DC) Hiern Relevant in the Management of Alzheimer’s Disease"

_antioxidants, 2019, doi:10.3390/antiox8090400_

Reviewer 1 Report

Introduction need to be revised.

The authors need to revise the following sentences:

Page 2, line 42: …. “from one form of neurodegenerative disease (ND) or the other”… Please be more clear.

Page 2, lines 42-43: …” expected rise in this figure”. … What figure?

Page 2, lines 45-47:However, several factors including aging [5] and some pathological conditions such as impaired mitochondrial function [6], aggregated proteins deposit [7], neuroinflammation [8], cholinergic deficit [9] and oxidative stress [10] have been associated with NDs”…. This sentence describes hallmarks od Alzheimer (AD), not all NDs. Because of the anticholinesterase activity, the authors should specifically refers to AD. Then, based on the antioxidant and scavenging activities, they can say that mitochondrial dysfunction and oxidative stress are relevant in all neurodegenerative conditions (NDs)…

Page 2, line 56: ……” It’s also widely    “…. The authors should avoid contractions in scientific texts

Materials and Methods need to be revised.

Page 2, line 64: ……authenticated at by Mr. Odewo”… “at by” is not correct.

Page 3, lines 74-75: Please briefly describe the standard procedures

Page 3, line 79:…followed by 0.5 mL”…. Do you mean: followed by addition of 0.5 mL

Page 3, line 82:… “total phenolic”…. Do you mean: total phenolic compounds?

Page 3, line 96:… “To 0.5ml of extract/fractions (1mg/ml) was added 3ml of 4% vanillin in methanol and 1.5 ml of hydrochloric acid (HCl).”…. Please, revise the construction of this sentence.

Page 3, line 105:… “and the positive controls (Ascorbic acid”….. Do you mean “OR” the positive controls…

Page 4, line 105:… “Substances which have reduction potential”…..  Please, revise

The authors should be consistent. Sometimes they use “minutes”, sometimes they use the abbreviation. They should always use the abbreviation

Results

Page 5, line 179:… “ (Table 2)”… Here, the authors start with Table 2, but a table 1 should be mentioned in the text before introducing table 2.

Page 5, line 179:… “polar solvent was able to extract more”… The adjective “More” needs to be followed by a noun. More what?

Page 6, line 182:… “(Table 1).”… Table numbering should be in sequential order.

Page 6, lines 194-199: “Phenolic compounds are present in…………….… that has also been implicated as antioxidant [36]”… .. This sentence for description of compounds should be better in the Introduction.

Page 7, line 203-204: “Data are expressed as Mean ± 203 SEM (n=3). GAE: Gallic Acid Equivalent, Methanol extract (ME), Hexane fraction (HF), Ethyl acetate fraction (EF), Aqueous fraction (AF)”… .. Is this a legend of table 2 ?? There is not appropriate spacing between text and legends. Moreover, figure legends should describe what is shown in the graphs, not just how data are expressed (mean and SEM)

Page 7, line 209-210: “DPPH is usually reduced by a hydrogen donating compound leading to its change in colour, from deep violet to light yellow, which can be monitored spectrophotometrically [37].”… .. This should be better in the corresponding Mat& method section.

Page 8, line 209-210: “At physiological 223 pH (7.2), sodium nitroprusside decomposes in aqueous solution to produce NO which reacts with oxygen to form stable products: nitrate and nitrite. Scavengers of NO compete with oxygen leading to reduced production of nitrite ions [40]”… .. This should be better in the corresponding Mat& method section.

Page 10, line 260: “….dual inhibitory effect on AChE and BuChE are also considered better”… .. Please, explain why.

Page 11, line 280: Table 4 is not mentioned in the main text.

Conclusions

Page 14, line 315-316: “This study revealed the potential use of M. longistyla in the management of neurodegenerative conditions” ….. This conclusion is not appropriate. All assay ar in-vitro, not even on cells. Therefore, the authors should say that “This study revealed the ……. activity of compounds present in M. longistyla and suggest the potential use of extracts from this plant for the management of neurodegenerative conditions....

The authors need to check the english

Author Response

Please seethe attachment.

Reviewer 2 Report

The authors of this manuscript have investigated the in vitro inhibitory properties of extracts obtained from leaves of Macrosphyra longistyla against the activity of acetylcholinesterase. Likewise, they have also investigated the antioxidant properties of those extracts in vitro. Their results show the existence of both kinds of properties in the studied extracts, locating the highest activity in the most polar fraction. Based on these results, the authors highlight throughout their manuscript the possible usefulness of those extracts for the treatment of Alzheimer's disease. The subject of this research is interesting, the methodology used is correct, and the manuscript is well-written. However, the manuscript is too speculative because the authors have not performed any in vivo experiment allowing them to assess the real capacity of these extracts in a living biological system. In addition, the fact that the maximum activity was in the polar fraction, does not allow foreseeing that these substances can easily cross the blood-brain barrier. This work can be published showing the here reported antioxidant properties and acetylcholinesterase inhibitory properties, but not proposing Macrosphyra longisty extracts as a potential treatment for Alzheimer's disease.

Author Response

Round  2

Reviewer 1 Report

The authors need to be more kind/polite in some responses to the reviewer.

If they do not like comments, then they should publish their work on NON-peer reviewed journals.

Reviewer 2 Report

This manuscript may be accepted for publication in its current form.